# Study on the Interaction between Serum Thyrotropin and Semen Parameters in Men

**DOI:** 10.3390/medsci10020022

**Published:** 2022-04-04

**Authors:** Ioannis Kakoulidis, Ioannis Ilias, Stefanos Stergiotis, Stefanos Togias, Aikaterini Michou, Anastasia Lekkou, Vasiliki Mastrodimou, Athina Pappa, Charalampos Milionis, Evangelia Venaki, Eftychia Koukkou

**Affiliations:** Department of Endocrinology, Diabetes and Metabolism, Elena Venizelou General and Maternity Hospital, GR-11521 Athens, Greece; i_kakoulidis@yahoo.gr (I.K.); stef_ster@hotmail.com (S.S.); s.tog90@gmail.com (S.T.); katerina.michoy@yahoo.com (A.M.); anastasia.lk@gmail.com (A.L.); si-li-a@hotmail.com (V.M.); athpappa@gmail.com (A.P.); pesscharis@hotmail.com (C.M.); evivenaki@gmail.com (E.V.); ekoukkou@gmail.com (E.K.)

**Keywords:** male, human, semen analysis, thyroid, thyrotropin

## Abstract

The effect of thyroid function on semen parameters has been studied in pathological conditions in small studies. With this research work, we aimed to study thyroid hormone effects on semen parameters in 130 men who were evaluated for couple subfertility. Our study was cross-sectional. We noted semen volume, sperm concentration, total sperm count, testosterone levels and thyrotropin (TSH) levels. The analysis included ordinary least squares regression (OLS-R), quantile regression (QR) and segmented line regression (SR). Using OLS-R, a weak negative correlation was found between the logTSH levels and semen volume (r = −0.16, r^2^ = 0.03, *p* = 0.05). In Q-R, each incremental unit increase in logTSH decreased the mean semen volume between −0.78 ± 0.44 and −1.33 ± 0.34 mL (40–60th response quantile) and between −1.19 ± 0.71 and −0.61 ± 0.31 mL (70–90th response quantile) (*p* = 0.049). With SR, a biphasic relationship of sperm concentration with TSH was noted (positive turning to negative, peaking at TSH = 1.22 μIU/mL). Thus, a weak negative association between the TSH levels and semen volume was noted, showing a trough within the usual normal range for TSH. Moreover, a biphasic relationship between the sperm concentration and TSH was also noted, peaking at approximately mid-normal TSH levels. Based on our results, TSH explained slightly less than 3% of the variation in semen volume and 7% of the sperm concentration (thus, other factors, which were not studied here, have a more important effect on it).

## 1. Introduction

The role of thyroid hormones is important in testicular development and function (with involvement in Sertoli cell proliferation and functional maturation control and postnatally in Leydig cell differentiation and steroidogenesis) [1,2,3,4]. Additionally, thyroid hormone receptors are found in testicular cells throughout development and in adulthood [2,5]. Thyroid hormones can interfere in both androgen biosynthesis and spermatogenesis, either through a direct action on Leydig and Sertoli cells or by modulating the secretion of gonadotropins [6]. However, the research evidence on the underlying physiologic processes is currently vague and non-consistent and mainly refers to non-primate species. Although the participation of thyroid hormones in the physiology of testes before and during puberty is well-known, its role in adult testes is still undefined. It possibly contributes to the regulation of sperm quality, but the cellular sequence of events is still largely uncovered. The two most common types of thyroid dysfunction include hyperthyroidism and hypothyroidism. Thyrotoxicosis may result in oligozoospermia, asthenozoospermia, and/or teratozoospermia and reduced semen volume. Most semen parameters usually revert to normal upon treatment of hyperthyroidism. Moreover, seminal vesicle volume and emptying and fructose concentration are positively correlated with the serum T3 levels. The most frequent semen abnormality that occurs in hypothyroidism is teratozoospermia. Decreased sperm motility and altered secretory activity of the sex-accessory glands with a low ejaculate volume have been also reported. Semen alterations during hypothyroidism are reversible after achieving euthyroidism [1]. Yet, the effect of thyroid function on semen parameters has been studied in pathological conditions in mostly small clinical studies [7]. Our aim was to study thyroid hormone effects on semen parameters in a larger sample using handy data from our current clinical practice.

## 2. Materials and Methods

We retrospectively analyzed anonymized data from a large series of outpatients (*n* = 677) who were evaluated for couple subfertility in our andrology clinic from 2007 to 2017 (cross-sectional study). From this initial series, we excluded men with incomplete data, thyroid or pituitary disorders, azoospermia (but not cryptozoospermia [8], with normal follicle-stimulating hormone (FSH) levels and, in most subjects, testicular biopsy), those who had surgery for varicocele, who had active infections (in the genital tract or elsewhere) or were receiving medications that altered testicular or thyroid physiology. A series of 130 men from the records of the clinic were selected (mean age ± SD: 34.3 ± 6.2 years, mean body mass index (BMI) ± standard deviation(SD): 27.9 ± 4.8 kg/m^2^). We noted the semen volume, sperm concentration (in ×10^6^/mL), total sperm count per the techniques and criteria set forth by the World Health Organization [8], plasma testosterone and thyrotropin (TSH) levels (using electrochemiluminescence assays; CobasElecsys, Roche, Basel, Switzerland). The selection of TSH was guided by its utility as the screening test of choice for thyroid function [9]. The statistical analysis of each variable (age, BMI, semen volume, sperm concentration, total sperm count and testosterone levels) versus TSH was done with ordinary least squares regression (OLS-R), quantile regression [10] (QR, both OLS-R and QR with Gretl, v.2019d; http://gretl.sourceforge.net/index.html (accessed on 3 February 2020)) and segmented line regression (SR, with SegReg v.2014, Oosterbaan R.J, International Institute for Land Reclamation and Improvement, Wageningen, The Netherlands; www.waterlog.info (accessed on 3 February 2020)). Whereas OLS-R estimates the conditional mean of the response variable against the values of the predictor variables, QR estimates the conditional median (or other quantiles) of the response variable and, as such, provides more robust estimates, particularly in the presence of outliers. In analysis with SR, the dataset is divided into bins at breakpoints, and each bin has its separate fit. Assessment of the semen parameters with TSH was also done with analysis of variance (ANOVA) [8]. A post-hoc power calculation followed the analyses [11,12]. This work followed the rules of the Declaration of Helsinki. It has been part of an ongoing project/study of male fertility for more than ten years in our institution, which was approved by the Elena Venizelou General and Maternity Hospital Institutional Review Board in 2009 (Approval No. 04/2009)—this was also stated in an older publication of our group [13].

## 3. Results

The mean testosterone was 4.41 ± 1.86 ng/mL, semen volume: 3.19 ± 1.44 mL and total sperm count: 48.8 ± 77.3 × 10^6^. Thirty-six subjects had sperm concentrations higher than 15 × 10^6^/mL (the accepted lower normal limit) (Table 1). The mean ± SD TSH levels were 1.58 ± 1.01 mIU/L; seven subjects had TSH <0.4 μIU/mL, and one had TSH >4.5 μIU/mL. The TSH levels, according to the total sperm count grouping, were not significant (ANOVA *p* > 0.5).

Using OLS-R, a weak negative correlation was found between the logTSH levels and the semen volume (r = −0.16, r^2^ =0.03, *p* = 0.05) (Figure 1).

Further analysis with Q-R showed that each incremental unit increase in logTSH decreased the mean semen volume between −0.78 ± 0.44 and −1.33 ± 0.34 mL (for the 40–60th response quantiles, i.e., semen volume) and between −1.19 ± 0.71 and −0.61 ± 0.31 mL (for the 70–90th response quantile) (*p* = 0.049) (Figure 2).

Assessment with SR of the sperm concentration versus logTSH showed a breakpoint at TSH = 1.22 μIU/mL (r = +0.24, r^2^ = 0.06, *p* = 0.07 for TSH ≤ 1.22 μIU/mL and r = −0.26, r^2^ = 0.07, *p* = 0.02 for TSH > 1.22 μIU/mL) (Figure 3).

No significant differences were noted between the semen parameters for subjects with TSH lower than 0.4 μIU/mL or higher than 4.5 μIU/mL versus subjects with normal TSH (data not shown for brevity). The post-hoc power analysis (for a probability level alpha of 5% or 10%) found an attained power of 24% or 36%, respectively.

## 4. Discussion

We noted a weak negative association between the TSH levels and semen volume, showing a trough within the usual normal range for TSH. Moreover, we noted a biphasic relationship of the sperm concentration with TSH, peaking at approximately mid-normal TSH levels.

Thyroid function is under control of the hypothalamic–pituitary–thyroid axis (HPT axis). TSH plays a pivotal role in the regulation of the HPT axis. Although TSH is released in a pulsatile rhythm, TSH and thyroid hormone excursions are modest mainly because of their long plasma half-life (50 min and 7 days, respectively). Thus, single measurements of TSH and thyroid hormones are usually adequate for assessing the HPT axis [14]. Testicular function is modulated by the hypothalamic–pituitary–gonadal axis (HPG axis) in adult men. Hypothalamic gonadotropin-releasing hormone (GnRH) regulates the secretion of the pituitary gonadotropins, namely FSH and luteinizing hormone (LH). LH acts primarily on Leydig cells to stimulate the production of testosterone, whilst FSH acts on the Sertoli cells to promote spermatogenesis and induce the synthesis of specific products such as inhibins and activins. Leydig and Sertoli cell functions are interrelated at several levels. Testosterone and estrogen exert negative feedback on both the hypothalamus and the pituitary, whilst inhibin selectively downregulates pituitary FSH [15]. There is a crosstalk between the thyroid function and HPG axis. The thyroid hormones include 3,5,3′-triiodothyronine (T3) and 3,5,3′,5′-tetraiodothyronine or thyroxine (T4). T4 is released from the thyroid gland in about twentyfold excess over T3, but most of T4 is converted to T3 by deiodinases type 1 and 2 in target tissues. The inactivation of T4 and T3 is accomplished by deiodination to reverse 3,3′,5-triiodothyronine (rT3) and 3,3′-diiodothyronine (T2), respectively. The cellular uptake of iodothyronines is mediated through passive diffusion and by specific transporter proteins, including monocarboxylate transporters 8 and 10 (MCT8 and MCT10) and organic anion-transporting polypeptide 1C1 (OATP1C1) [16]. Thyroid hormones exert their biological effects by binding to specific nuclear thyroid hormone receptors (TRs). There are two different TRs: TRα and THRβ. Both receptors are distributed variably among organs. TRα is abundant in the brain, kidneys, gonads, muscle and heart, whilst TRβ predominate in the pituitary gland and liver. Activated TRs bind to thyroid hormone response elements (TREs), located in the promoter region of target genes, and activate or repress the transcription of DNA. TRs are widely present in the different compartments of the testes and the sex-accessory glands [17]. Testosterone, FSH, LH and other hormones, such as activin and follistatin, influence spermatogenesis [18,19]. Disturbances at any stage could potentially lead to defective sperm production. Sertoli cells are somatic cells in the seminiferous tubules of the testes. Their main function is to nurture the developing sperm cells through the stages of spermatogenesis via direct action and by controlling the environment within the seminiferous tubules. The regulation of spermatogenesis is mediated by the action of FSH and testosterone on Sertoli cells [20]. Two biologically active TR isoforms are present in Sertoli cells: TRα1 and TRβ1 [21]. T3 inhibits Sertoli cell proliferation and stimulates their differentiation in prepubertal testes, thereby having a central role in the transition from an immature to a functionally mature Sertoli cell [22]. Thyroid hormones also play a critical role in the onset of Leydig cell differentiation before puberty [23]. T3 binds to TRs on post-pubertal Sertoli cells and downregulates the synthesis of aromatase and androgen-binding protein. In this way, it modulates the conversion of testosterone to 17β-estradiol and the concentration of testosterone in the seminiferous tubules. In the adult Leydig cells, thyroid hormones promote the steroidogenesis of testosterone through the stimulation of the steroidogenic acute regulatory (StAR) protein that is involved with cholesterol transport into the mitochondria. TRs have also been identified in germ cells during the various stages of spermatogenesis, but the potential biological role of the thyroid hormone in these cells remains to be elucidated [23]. Proper spermatogenesis also depends on the modulation of methylation that occurs in undifferentiated sperm cells, such as spermatogonia [24]. The methylation of genes that are implicated in DNA repair during meiosis in spermatogenesis are associated with Reactive Oxygen Species (ROS) [25]. ROS are byproducts of metabolism in aerobic cells that are normally treated by physiological antioxidants. Oxidative stress is a consequence of an imbalance between the production of ROS and their elimination by the body’s antioxidant arsenal. Oxidative stress is well-known to be detrimental to sperm quality mainly by damaging DNA in sperm cells. The physiological ranges of thyroid hormones regulate metabolism, including oxygen consumption by cells, and both hyperthyroidism and hypothyroidism are associated with oxidative damage. Although the role of thyroid hormones in regulating oxidative stress specifically in male reproductive organs is currently not clear, it is inferred that oxidative parameters are stabilized in a euthyroid state [26]. A low semen volume was reported in a small series (*n* = 10–25) of overtly hyperthyroid men [27,28]; in our study no subject was overtly hyperthyroid, although a few subjects had very low TSH. Not surprisingly, in our subjects, the relationship of TSH with the semen parameters was more accentuated around the middle of the normal range of TSH values. In small (*n* = 24–25) studies of hypothyroid subjects, lower sperm counts compared to controls have been reported with no precise mention of the semen volume [29,30]. Our results lend, in part, credence to those of Lotti et al. [31], who found that the euthyroid subjects (*n* = 145) had lower semen volumes compared to those with subclinical hyperthyroidism (*n* = 6) and higher than those with subclinical hypothyroidism (*n* = 12); however, no association of the semen parameters with TSH was noted. In a large (*n* = 1098) study from China, suboptimal or excessive dietary iodine intake was associated with a slightly lower semen volume compared to the optimal dietary iodine intake; however, the authors did not present the thyroid function status [32]. The most prominent effect of thyroid function abnormalities on semen parameters seems to be on impaired motility [33,34,35] and/or morphology [36].

This study has the advantage of examining mostly subjects with normal TSH and inferred normal thyroid function. The quantitative effect of TSH on semen appeared to be biphasic, with a slight downward inflection in the mid-normal region of values for volume and TSH and an upward inflection in the mid-normal region of values for sperm concentrations and TSHs. Based on our results, logTSH explained slightly less than 3% of the variation in semen volume, and TSH explained less than 7% of the variations in sperm concentration (thus, other factors, which were not studied here, have a more important effect on it). The semen volume and sperm concentration are important, since they are taken into consideration for the assessment of the sperm quality [7]. However, we have to point out the limitations of the present study. The assessments were rudimentary: apart from TSH, no thyroid hormones were measured per se, and the included semen parameters were not complete (form and motility were not available for analysis).

Based on our results, we believe that the TSH–semen volume/sperm concentration relationship can be stronger (the calculated post-hoc power was low, though most relevant studies in the literature had fewer subjects than the present one). Thyroid hormones are now recognized to have a drastic role in the male gonadal development and reproductive function, although the relevant mechanisms are not fully understood. They probably act via genomic and nongenomic mechanisms on various vital cells of the male reproductive system and regulate the testicular secretion of testosterone and the concentration of seminal plasma components (calcium, fructose, magnesium, zinc, etc.). The appropriate amount of intratesticular testosterone promotes spermatogenesis, whilst other seminal properties facilitate sperm motility and viability and maintain the semen volume [23]. Therefore, a proper understanding of the relationship between thyroid function and sperm quality is essential for the development and implementation of effective preventive and therapeutic interventions against male subfertility. For this purpose, extensive research is needed to unveil the role of the thyroid gland and its disorders in the preservation and deterioration of male reproductive health and to draw more reliable conclusions that would be useful in clinical practice.

In conclusion, in this study, we found weak associations between the TSH and semen volume and concentration. As such, these findings do implicate normal thyroid function as an aspect of good semen quality but also point to the existence of other, stronger factors that influence semen quality.

## Figures and Tables

**Figure 1 medsci-10-00022-f001:**
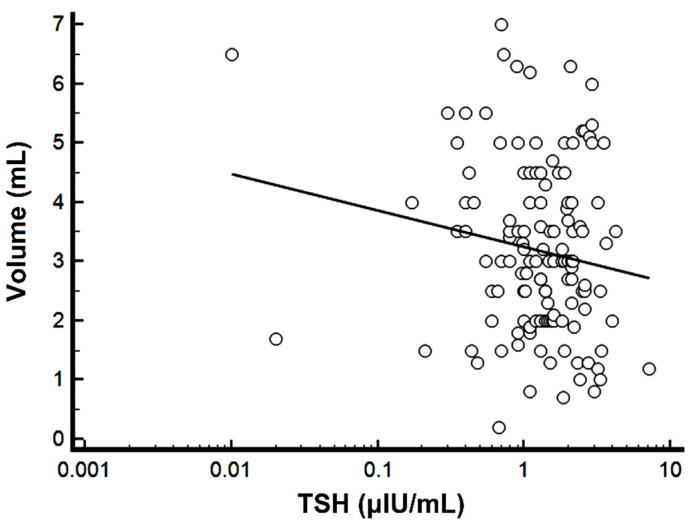
Scatter diagram of the semen volume versus TSH. Note the logarithmic scale for TSH.

**Figure 2 medsci-10-00022-f002:**
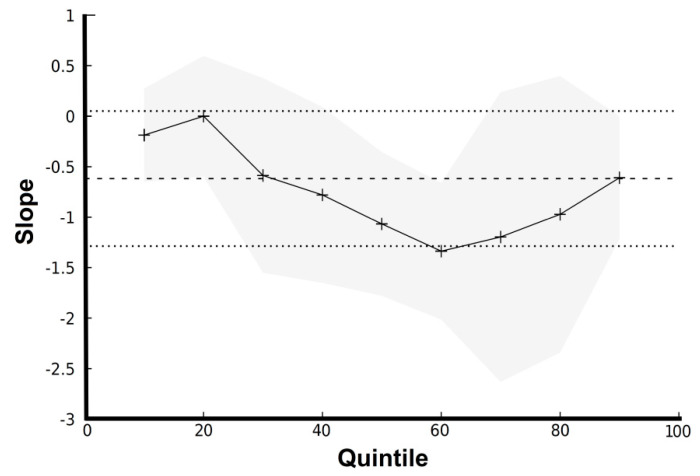
Graph of slopes for quintiles of semen volume versus logTSH. Note that the 50th quantile corresponds to the median quantile of the response (i.e., semen volume). The straight line with dashed line borders indicate the OLS-R estimates with 95% confidence intervals, whereas the splined line with grey borders indicates the Q-R estimates with 95% confidence intervals.

**Figure 3 medsci-10-00022-f003:**
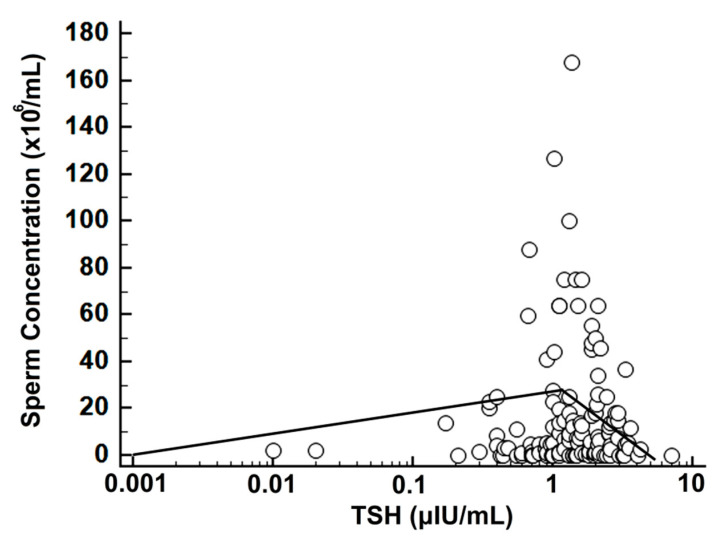
Scatterplot of logTSH versus sperm concentration with SR; note the logarithmic scale for TSH and the biphasic regression line with a breakpoint at TSH = 1.22 μIU/mL.

**Table 1 medsci-10-00022-t001:** Total sperm count of the subjects.

Total Sperm Count	<5 × 10^6^	5–19 × 10^6^	20–39 × 10^6^	≥40 × 10^6^
*n*	43	30	16	41
TSH in μIU/mL (mean ± SD) *	1.58 ± 1.23	1.59 ± 1.14	1.69 ± 0.79	1.53 ± 0.76

* The laboratory’s normal range is >0.5 μIU/mL to <4.5 μIU/mL.

## Data Availability

The data for this study are available at https://doi.org/10.5281/zenodo.6410224 (accessed on 3 February 2020).

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
