# Peer review of "Study on the Interaction between Serum Thyrotropin and Semen Parameters in Men"

_medsci, 2022, doi:10.3390/medsci10020022_

Round 1

Reviewer 1 Report

The study reports the relationship among serum thyrotopin levels and semen parameters. It is very interesting but it lacks some important revision as listed bellow:

  1. Tittle - I believe that this is not a completely informative tittle. A suggestion: Study on the interactions between serum thyrotopin and semen parameters in men
  2.  Abstract- Are the words: background, aim, results, etc really necessary? Couldn't the text simply be more fluid and direct without these words? Moreover, authors should present some values for sperm parameters from fertile vs subfertile men affected by tyroid pathologies.
  3. Keywords - ok
  4. Introduction is extremely reduced. With basis on current literature, authors should emphasize the importance of their study for the current knowledge on the subject. What is the knowledge gap that the study intends to fill? How does it differ from existing studies? The aim is so simplistic. Please rewrite it.
  5. Material and Methods - It is so summarized. Authors should present them in separate topics. What was the average age of the men used? How each semen parameter was evaluated? Please detail the statistical methods used for the analysis.
  6. Results - If different sperm concentrations determined different groups of men, why other parameters were not presented per group? Other results should be rewritten for more clarity to the reader.
  7. Discussion - It brings some interesting information but authors should use literature information to try to explain the results found. Lines 97 to 179 are merely literature review;  despite the very important information, it is not written with a link to the results found and it should be rewritten. Finally, a final paragraph clearly presenting the main conclusions would be welcome.

Author Response

We thank Reviewer #1 for the time and effort spent on assessing our manuscript [“medsci-1654320”].

Reviewer #1

 [1]. Title - I believe that this is not a completely informative title. A suggestion: Study on the interactions between serum thyrotopin and semen parameters in men

In the Revised version of the Manuscript the title has been changed accordingly

[2].  Abstract- Are the words: background, aim, results, etc really necessary? Couldn't the text simply be more fluid and direct without these words? Moreover, authors should present some values for sperm parameters from fertile vs subfertile men affected by thyroid pathologies.

In the Revised version of the Manuscript we eliminated the words which were indicated by the Reviewer. Further additions to the abstract however were avoided given the limited word count per the Journal’s instructions to Authors.

[3]. Introduction is extremely reduced. With basis on current literature, authors should emphasize the importance of their study for the current knowledge on the subject. What is the knowledge gap that the study intends to fill? How does it differ from existing studies? The aim is so simplistic. Please rewrite it.

The Introduction has been expanded in the Revised version of the Manuscript as follows: “The role of thyroid hormones is important in testicular development and function (with involvement in Sertoli cell proliferation and functional maturation control and postnatally in Leydig cell differentiation and steroidogenesis) [1-4]. Additionally, thyroid hormone receptors are found in testicular cells throughout development and in adulthood [2,5]. Thyroid hormone can interfere in both androgen biosynthesis and spermatogenesis either through a direct action on Leydig and Sertoli cells or by modulating the secretion of gonadotropins [6]. However, the research evidence on underlying physiologic processes is currently vague and non-consistent, and mainly refers to non-primate species. Although the participation of thyroid hormone in the physiology of testis before and during puberty is well known, its role in the adult testis is still undefined. It possibly contributes to the regulation of sperm quality, but the cellular sequence of events is still largely uncovered. The two most common types of thyroid dysfunction include hyperhyroidism and hypothyroidism. Thyrotoxicosis may result in oligozoospermia, asthenozoospermia, and/or teratozoospermia and reduced semen volume. Semen parameters most usually revert to normality upon treatment of hyperthyroidism. Moreover, seminal vesicle volume and emptying and fructose concentration is positively correlated with serum T3 levels. The most frequent semen abnormality that occurs in hypothyroidism is teratozoospermia. Decreased sperm motility and altered secretory activity of the sex-accessory glands with low ejaculate volume have been also reported. Semen alterations during hypothyroidism are reversible after achieving euthyroidism [1]. Yet, the effect of thyroid function on semen parameters has been studied in pathological conditions in mostly small clinical studies [7]. Our aim was to study thyroid hormone effects on semen parameters in a larger sample using handy data from our current clinical practice.”

[3]. Material and Methods - It is so summarized. Authors should present them in separate topics. What was the average age of the men used? How each semen parameter was evaluated? Please detail the statistical methods used for the analysis.

The age of the men studied was given in this section: “mean age±SD: 34.3±6.2 years”. In the revised version of the manuscript we state that “The semen parameters were evaluated per the techniques and criteria set forth by the World health Organization”. The statistical methods are further clarified by the addition of the following: “Whereas [ordinary least squares regression] OLS-R estimates the conditional mean of the response variable against the values of the predictor variables, [quintile regression] QR estimates the conditional median (or other quantiles) of the response variable, and as such provides more robust estimates, particularly in the presence of outliers. In analysis with [segmented line regression] SR, the dataset is divided into bins at breakpoints and each bin has its separate fit.”

[4]. Results - If different sperm concentrations determined different groups of men, why other parameters were not presented per group? Other results should be rewritten for more clarity to the reader.

In the Revised version of the Manuscript a new Table has been provided and new results were added.

[5]. Discussion - It brings some interesting information but authors should use literature information to try to explain the results found. Lines 97 to 179 are merely literature review;  despite the very important information, it is not written with a link to the results found and it should be rewritten. Finally, a final paragraph clearly presenting the main conclusions would be welcome.

In the Revised version of the manuscript the text that the Reviewer refers to has been shortened and a Conclusion has been appended.

Reviewer 2 Report

The manuscript (ID:  medsci-1654320) entitled “Study of serum thyrotropin versus semen parameters” by Dr. Kakoulidis and colleagues is a cross-sectional study reporting on an investigation of the relationship between testosterone levels and thyrotropin (TSH) levels on semen volume, sperm concentration and total sperm count in a group of n=130 individuals. The work is potentially interesting. However, several improvements are necessary to be made. The main weakness of the manuscript is represented by the unbalance between the introductive section and the discussion. The introduction should be improved by including the state of the art of the field, mainly the thyroid hormones and their role in testicular development and function as well as in impacting semen parameters.
In summary, I recommend a major revision. I have several observations: 

Major comments
1.    As mentioned before, the introductive section should be implemented with additional notions on the role of thyroid hormones and their role in testicular development and function as well as in impacting semen parameters. Several parts in the discussion can be moved to the introduction, especially those describing the HPT axis function. 
2.    Have pathogenic infections been considered as exclusion criteria?
3.    The main findings of the work should be more deeply discussed in the context of previous findings reported in the literature
4.    Since semen volume, sperm concentration and total sperm count being investigated, the sperm morphology should also be included in the analysis. Alternatively, the lack of morphological data should be included in the discussion in the study limitations section.
5.    A brief conclusion should be included at the end of the discussion 

Minor 
Line 13 If not necessary, “Aim:” “ results” ets.. should be removed as superfluous
Line 136 besides hormonal regulation, proper spermatogenesis also depends on methylation modulations occurring in undifferentiated sperm cells, such as spermatogonia (described in detail here doi.org/10.3389/fcell.2021.689624). This important information and reference should be included

Author Response

We thank Reviewer #2 for the time and effort spent on assessing our manuscript [“medsci-1654320”].

Reviewer #2

[1].    As mentioned before, the introductive section should be implemented with additional notions on the role of thyroid hormones and their role in testicular development and function as well as in impacting semen parameters. Several parts in the discussion can be moved to the introduction, especially those describing the HPT axis function.

The Reviewer is absolutely right and we have followed his/her advice, moving parts of the Discussion to the paper’s Introduction.

[2].    Have pathogenic infections been considered as exclusion criteria?

Men with infections were excluded; this has been clarified in the Methods Section of the Revised version of the Manuscript.

[3].    The main findings of the work should be more deeply discussed in the context of previous findings reported in the literature

In the Revised version of the Manuscript we have streamlined the Discussion section, by moving and/or eliminating parts of it, honing on the literature and adding a very brief concluding remark.

[4].    Since semen volume, sperm concentration and total sperm count being investigated, the sperm morphology should also be included in the analysis. Alternatively, the lack of morphological data should be included in the discussion in the study limitations section.

Sperm morphology/form was not available for this work; this is indicated in the limitations of the study

[5].    A brief conclusion should be included at the end of the discussion .

This has been added in the Revised version of the manuscript.

[6]. Line 13 If not necessary, “Aim:” “ results” ets.. should be removed as superfluous

This has been done in the Revised version of the Manuscript

[7]. Line 136 besides hormonal regulation, proper spermatogenesis also depends on methylation modulations occurring in undifferentiated sperm cells, such as spermatogonia (described in detail here doi.org/10.3389/fcell.2021.689624). This important information and reference should be included.

This pertinent detail has been incorporated in the Revised version of the manuscript, also adding the suggested reference.

Round 2

Reviewer 1 Report

Authors adressed all the required revisions in this new version. To the best of my knowledge, manuscript could be accepted for publication at the present form. 

Reviewer 2 Report

The Authors have addressed all of my concerns with the original manuscript. The revised manuscript is ready for publication.